# Wearables in Nephrology: Fanciful Gadgetry or Prêt-à-Porter?

**DOI:** 10.3390/s23031361

**Published:** 2023-01-26

**Authors:** Madelena Stauss, Htay Htay, Jeroen P. Kooman, Thomas Lindsay, Alexander Woywodt

**Affiliations:** 1Department of Nephrology, Lancashire Teaching Hospitals NHS Foundation Trust, Preston PR2 9HT, UK; 2Department of Renal Medicine, Singapore General Hospital, Singapore 169608, Singapore; 3Department of Internal Medicine, Division of Nephrology, Maastricht University, 6229 HX Maastricht, The Netherlands

**Keywords:** wearables, telemedicine, nephrology, chronic kidney disease, dialysis, transplantation

## Abstract

Telemedicine and digitalised healthcare have recently seen exponential growth, led, in part, by increasing efforts to improve patient flexibility and autonomy, as well as drivers from financial austerity and concerns over climate change. Nephrology is no exception, and daily innovations are underway to provide digitalised alternatives to current models of healthcare provision. Wearable technology already exists commercially, and advances in nanotechnology and miniaturisation mean interest is also garnering clinically. Here, we outline the current existing wearable technology pertaining to the diagnosis and monitoring of patients with a spectrum of kidney disease, give an overview of wearable dialysis technology, and explore wearables that do not yet exist but would be of great interest. Finally, we discuss challenges and potential pitfalls with utilising wearable technology and the factors associated with successful implementation.

## 1. Introduction

With a global prevalence of 9.1% and the 12th leading cause of death, chronic kidney disease (CKD) contributes substantially to morbidity and mortality [1]. The prevalence of CKD and kidney replacement therapy (KRT) through dialysis and transplantation continues to rise, but CKD-related mortality has not seen the continued improvement that other long-term conditions such as cancer, lung disease or cardiovascular disease have [1]. 

Nephrology has viewed itself as a technological speciality ever since the early days when Willem Kolff in the 1940s first saved a patient’s life using dialysis equipment he had assembled from a car radiator, sausage casing and parts of a downed fighter aircraft. Since then, the specialty has expanded its focus beyond dialysis but retained the interest in technology and innovation. Not surprisingly, telemedicine has seen exponential growth in nephrology, in part driven by the COVID-19 pandemic [2] but also as a result of financial austerity and worsening climate change. Patients and relatives’ lifestyles and priorities are also constantly evolving, leading to a constant quest for innovative models of flexible patient-centred care.

Within nephrology, point-of-care testing (POCT) at home has recently received significant interest for a variety of applications [3,4,5,6]. However, none of these approaches currently exist in wearable form. Here, we provide a brief review of wearable technologies in nephrology that are currently available or expected in the very near future. We also describe unmet needs in terms of technologies that are desirable from the clinician point of view but do not currently exist. Finally, we consider pitfalls and challenges to using wearable technology and implications for resources, workforce and training.

## 2. Wearable Devices for Diagnosis and Monitoring in CKD

Wearable or mobile sensors have entered healthcare both as medically approved devices and through the consumer market, and interest in medical applications of wearable sensors has grown further since the COVID-19 pandemic [7]. The increasing availability of wearable sensors brings both opportunities and challenges. On the positive side, deteriorations in health conditions may be detected earlier, which may prompt pre-emptive interventions. Moreover, reliable and actionable data can increase patient empowerment and support shared decision making [8]. On the other hand, uncontrolled use of wearable technology might lead to false-positive results, followed by patient anxiety and unnecessary interventions. In addition, when not part of an integrated healthcare platform [9], they can lead to data overload and overburden healthcare providers [10]. Wearable or mobile sensors therefore can be, but are not necessarily, part of a remote healthcare monitoring and telehealth system [11].

Wearable sensors for patients with CKD do not represent a single class but can be categorized according to the underlying technology, type of device, timing of monitoring, clinical indication and the arena in which its use is authorised and approved (Figure 1). At present, much of the data on wearable technology focuses on patients with end-stage kidney disease (ESKD) on dialysis. With these patients, wearable sensors show great promise for monitoring during the inter- and intradialytic periods, in particular for patients on home-based dialysis [12]. In comparison, data on the use of wearable devices in the care of patients with CKD who are not on dialysis remain scarce and are largely extrapolated from devices developed from cardiovascular, electrolyte and physical activity perspectives but not studied specifically in the CKD cohort.

### 2.1. Cardiovascular Parameters in CKD

An important potential application of wearable technology is the detection of arrhythmias. Arrhythmias are common in patients with ESKD and related to outcome. Data from implantable loop recorders showed a high prevalence of atrial fibrillation (AF) but also stressed the importance of bradyarrhythmias as potential harbingers of sudden cardiac death in this population [13,14]. To the best of our knowledge, there are no published studies on arrhythmia detection using non-invasive, wearable devices specifically in this population. In addition to patches that have been specifically designed for this purpose, smartwatch and smartphone applications have also been designed for arrhythmia detection. Artificial intelligence algorithms applied to commercially available smartwatches have been shown to successfully detect left ventricular dysfunction [15]; which, in patients with CKD, may help to guide therapy or detect acute cardiac events in this high-risk population.

The principle of heart rate detection in these methods is based on photoplethysmography (PPG), which assesses volume changes in the wrist or finger due to changes in tissue perfusion [10]. Rhythm detection algorithms are based on PPG signals that are processed by machine learning in order to be translated into an arrhythmia signal [16]. Several of these algorithms, either using smartwatch- or smartphone-derived signals, have already been validated in populations without ESKD [17,18,19]. Furthermore, one smartwatch has the additional functionality of performing a single lead electrocardiogram (ECG), which improved the accuracy over its PPG-based algorithm [18]. However, it remains to be determined whether the additional detection of AF is clinically relevant in patients with ESKD, as consensus on the appropriateness and method of anticoagulant treatment remains unclear in this population [20]. For the detection of bradyarrhythmias using smartwatch-derived signals, only anecdotal reports are available and, again, not in the CKD or ESKD populations [21].

Blood pressure (BP) monitoring is an integral part of nephrology care either through office or home measurements or via ambulatory monitoring [22]. The early detection of hypotensive episodes could help prevent complications such as dizziness, falls and intradialytic complications in patients on home dialysis. Continuous home BP monitoring allows for the monitoring of a true average BP, as well as highlighting variations within the day and night. It is, however, less well tolerated than intermittent readings, likely due to its intrusive nature, particularly at night [23]. Recent developments include cuffless monitors and PPG-based techniques that may provide better-tolerated alternatives [24]. The latter could be incorporated into existing smartwatch technology, as it utilises either the pulse transit time or pulse arrival time derived from PPG and ECG signals [25]. Importantly, however, vascular abnormalities are common in CKD patients; not only due to traditional risk factors such as diabetes and hypertension but also more unique vascular insults such as calcium-phosphate product deposition, vasculitis and the presence of arteriovenous fistulae. The validity of PPG-derived techniques in these situations needs to be urgently clarified.

### 2.2. Biochemistry and Electrolytes in CKD

The monitoring of serum electrolytes is a large part of CKD management, and various techniques are in development that may provide a wearable option. Outside of CKD, wearable electrochemical biosensors exist in various forms capable of detecting cortisol, neuropeptides, ammonium and cytokines [26,27], as well as various pharmacological (e.g., levo-dopamine and penicillin), licit (e.g., caffeine and alcohol) and illicit (e.g., cocaine and opiates) drugs [28]. Urinary electrolyte analysis has been suggested as a potential avenue for the use of wearable technology [29], and a diaper-style sensor prototype has been developed that is capable of detecting urine [30]. However, this would clearly be unacceptable to the vast majority. An oral retainer has also been developed that can monitor either dietary sodium intake [31] or saliva uric acid levels [32], proffered by the authors as being used to guide dietary advice for hypertension management or urate-lowering therapies, respectively, both of which are pertinent to CKD. Again, however, compliance and tolerability are likely to limit the use of these products.

Non-invasive, wearable, flexible circuit board devices capable of monitoring real-time pH and calcium in sweat, urine or tears have been developed [33], which is pertinent to not only those with CKD but also in renal stone disease and hyperparathyroidism. However, sweat induced physiologically by exercise does not correlate reliably to either serum pH or numerous electrolytes, and sweat induced pharmacologically by pilocarpine iontophoresis may correlate with potassium only [34,35]. Concerns about skin integrity and discomfort with prolonged use have been raised as well [26]. Yang et al. devised an implantable, subcutaneous device capable of monitoring urea via piezoelectric-biosensing, which circumvents the issue of sweat–serum correlation [36].

Measuring electrolyte concentrations of interstitial fluid (ISF) is garnering much interest, akin to continuous glucose monitoring. The current methods of accessing ISF use either reverse iontophoresis, whereby ions migrate to the skin surface following stimulation by a mild electrical current (electrochemical method), or microneedling, where the ISF is directly accessed by puncturing the epidermis (physical method) [37]. A wearable patch is the most common approach, and various devices are in ex vivo proof-of-concept stages of testing animal skin for both sodium and potassium [38,39]. Correlation with the serum levels and in vivo testing is still required, and the variability of ISF penetration of solutes in different populations needs to be considered.

Finally, surrogate markers of electrolyte disturbances are also being trialled, as opposed to direct measurements of the electrolyte itself. Machine learning via deep convolutional neural networks has been shown to detect hyperkalaemia from a two-lead ECG [40], and more recently, a single-lead ECG trace successfully detected hyperkalaemia in which more than half of the studied patients had CKD [41]. As already discussed, current smartwatches can provide a single-lead ECG trace; therefore, it is conceivable that this technology could be integrated into existing wearable technology. Measuring a surrogate marker may hold the additional benefit of detecting clinically relevant deviations rather than absolute values.

### 2.3. Physical Activity in CKD

Most other evidence on wearable devices in patients with ESKD centres around the use of wearables for monitoring physical activity. In a systematic review, the mean step count was 4111 in patients on haemodialysis and 4264 in patients on peritoneal dialysis, clearly below the daily recommended target [42]. Reduced physical activity is related to mortality in this population [43], and recent studies have shown that a home-based exercise program is not only feasible [44] but also beneficial [45]. Although it is not yet known if it is acceptable to patients to wear activity monitors for prolonged periods of time, highlighting information gleaned from wearable technology may help with patient insight, motivation and engagement.

## 3. Wearable Dialysis Devices

The incidence of ESKD is increasing worldwide [46,47], and despite increases in kidney transplantation, the majority remain on dialysis. Dialysis is provided either via haemodialysis (HD), which can be in-centre or at home, or peritoneal dialysis (PD), which is typically delivered at home. Despite showing improved or comparable outcomes, home-based dialysis is underutilised and patient-related factors are often quoted as barriers [48]. One drawback of current dialysis provision is that dialysis machines are bulky, which hinders mobility, and the space required for machinery means that, for some patients, home-based therapy may not be possible. Furthermore, the lack of portability demands that patients spend lengthy hours attached to stationary dialysis machines and, in the case of in-centre HD, at fixed times, both of which restrict patient activities.

A wearable miniature dialysis machine would allow patients to dialyse anytime and anywhere and may provide freedom, enhance employability, increase autonomy, reduce the treatment burden and improve the quality of life of patients with ESKD. It has been suggested that clinical outcomes could also be affected. For HD, more continuous dialysis may lead to more stable solute clearance and ultrafiltration (UF), avoiding potentially unstable interdialytic fluid and electrolyte shifts, and for PD, the avoidance of higher glucose concentration solutions may help preserve the peritoneal membrane, and less frequent connects/disconnects reduce the risk of PD peritonitis—both causes of modality failure [49,50]. A wearable dialysis system requires the capacity to remove uraemic toxins, regenerate fresh dialysate continuously, maintain an acid–base balance and obtain adequate UF. It must also be safe, acceptable and easy to use for the patient.

There are both wearable PD [51] and HD devices in development, with some tested in animal studies and preliminary trials with humans. Both modalities require a fluid source; for PD, this is in the form of dialysate bags (typically 8–12 L/therapy), and HD is in the form of dialysate fluid plus water (typically 120–500 L/therapy, depending on the settings) [49]. Wearable dialysis therefore relies on the generation of fluid without connection to an external source. Simply speaking, the majority of wearable dialysis systems use a sorbent-based system whereby urea in the spent dialysis effluent is broken down (typically either enzymatically or through activated carbon [49]), then a series of sorbents either ab- or adsorb molecules via saturation or exchange mechanisms until an acceptable composition is reached [50]. The altered effluent then undergoes refreshing through the addition of bicarbonate and other electrolytes, plus glucose in the case of PD, before a series of further safety checks and returns to the body [50]. Urea removal via electrooxidation has also been explored; however, is currently not suitable due to safety concerns [52].

### 3.1. Peritoneal Dialysis

The Vicenza Wearable Artificial Kidney (ViWAK) PD system utilises a combination of different sorbents, including polystyrene resin and activated charcoal, to regenerate fresh dialysate [53]. It requires a double-lumen catheter for continuous flow, and in vitro experiments have shown a reduction in creatinine, β_2_-microglobulin and angiogenin over 10 h of therapy [53]. Conversely, the wearable artificial kidney Renart-PD system utilises electrochemical methods and a sorption purification system with activated carbon Kausorb-212 as a sorbent, and in bench testing has been shown to reduce urea, creatinine, uric acid and phosphorus, as well as stabilise pH [54]. The device is composed of a dialyser that separates spent dialysate with low molecular weight uraemic toxins in the regeneration circuit, where the removal of uraemic toxins by activated carbon and electrolysis takes place. The system can be operated by its control unit, a mobile application or a computer with special software [54]. However, to date, neither the ViWAK nor Renart-PD systems have published data on use in humans.

The Carry Life^®^ System PD is another wearable PD device that utilises an adsorbent system to remove uraemic toxins and adds glucose to the recirculating fluid to maintain a constant osmotic gradient in the peritoneal cavity [55]. It requires two catheters for dialysate to flow in and out of the peritoneal cavity simultaneously, and an in vivo study of four patients reported that 8-h therapy significantly reduced the serum creatinine, potassium and phosphate levels [55]. The same group has also developed the Carry Life^®^ System UF, which aims to achieve peritoneal UF. A study of five patients treated for eight hours found stable intraperitoneal glucose concentrations, UF was achieved and the treatment tolerated well [56].

Another wearable system with preliminary safety studies in humans is the Automated Wearable Artificial Kidney (AWAK) PD device (Figure 2), which is based on the REDY sorbent-based adsorption system together with the regeneration of fresh dialysate [57]. The AWAK device uses a tidal PD system that requires 250 mL of dialysate to move in and out of the peritoneal cavity continuously at a rate of 2 L/hour using a single lumen PD catheter (Figure 3) and weighs < 2 kg [57]. A study of 15 patients over three days demonstrated no serious adverse effects in any participant and found a significant reduction in the serum urea, creatinine, phosphate and β_2_-microglobulin [57]. However, there were some technical issues encountered in the early phase of the study relating to UF and adverse effects, particularly abdominal discomfort. After the preliminary study, the AWAK system underwent device modification with subsequent testing in animals and demonstrated improved UF in a porcine model [58]. The AWAK team is planning to conduct a single-arm feasibility study in PD patients.

Finally, the WEarable Artificial KIDney (WEAKID) is another sorbent-assisted device in development; however, it uses continuous flow PD technology [59]. The system circulates dialysate continuously in the peritoneal cavity using a standard PD catheter, removes toxins and generates fresh dialysate by a sorbent purification unit and also has the capacity for a gradual release of glucose [59]. In an animal study, it was reported to be safe and improved the clearance of small uraemic solutes [59]. The device has sensors for safety and automated control, a remote monitoring system and includes wearable and portable devices [60]. However, the portable device weighs approximately 12 kg and is intended for overnight use, and the wearable device weighs approximately 2.3 kg and is optional for daytime use if required. Therefore, the proposed therapy regime is not dissimilar to a current overnight automated PD regime with a manual daytime dwell, asking the question if this is truly a wearable option. The WEAKID group is planning to clinically validate the system in a feasibility human trial.

### 3.2. Haemodialysis

Fewer systems are available for wearable HD machines, although trials are currently underway. Safe and effective vascular access has thus far been a large barrier to wearable HD devices, with recent conceptualisations hoping to bridge this gap [61], in addition to the dilemma of safely anticoagulating the extracorporeal circuit [49,50]. Furthermore, cardiovascular stability must be considered, and although deep learning algorithms have been used to predict intradialytic hypotension in ESKD patients on HD [62], this was done with variables gleaned from a conventional dialysis machine that are not yet available on wearable technology.

The most studied prototypes are the Wearable Artificial Kidney (WAK) and Wearable Ultrafiltration (WUF) devices [63,64,65,66]. The early human trials of 24-h WAK therapy reported some technical issues, including clotting of the extracorporeal circuit, carbon dioxide bubbles in the dialysis circuit and variable blood and dialysate flow rates; however, there were no adverse cardiovascular events or acid–base and electrolyte disturbances [66]. This was the only trial to capture patient satisfaction data and reported higher satisfaction scores compared to in-centre HD [66]. With the initial system weighing around 4.5 kg [65], the WAK 3.0 HD system is smaller and lighter than the previous two prototypes and is currently in the trial stage.

Finally, MiniKid, a miniature artificial kidney, also utilises a sorbents-based purification system to generate fresh dialysate continuously [60]. The system requires a minimal amount of dialysate to operate; however, to date, no clinical study data are available.

In summary, various wearable dialysis systems are in differing stages of development. Technical challenges for both modalities still need to be rectified. Furthermore, although advances in miniaturisation and nanotechnology have meant devices are becoming smaller, some still weigh a substantial amount without including the weight of any dialysate fluid (typically 2–3 L) and changeable sorbent cartridges and supplies.

## 4. Devices That May Improve Patient Care but Don’t Yet Exist

### 4.1. Physiological Biometrics

Fluid overload is a common and preventable reason for hospital admission, morbidity and mortality [67], and fluid status is difficult to assess during virtual clinic appointments, often necessitating face-to-face reviews [68]. Measurements of weight and BP alone are not sensitive for the early detection of fluid overload [69]; however, remote biometric monitoring for PD patients has been shown to lead to more frequent dialysis prescription changes, less in person visits and an overall improved technique survival [70]. A wearable device that could carry out automated and objective assessments on a regular basis and thereby help guide decisions on dialysis parameters, dry weight and diuresis is therefore highly desirable.

One approach is to measure limb circumference as a surrogate for peripheral oedema and total body water volume. The “smart sock” currently in development does exactly this by incorporating multiple sensors in a sock [71]. Although simple, relatively inexpensive and completely non-invasive, this approach has multiple pitfalls, including the variable impact of exercise and posture on oedema distribution and the compression effect caused by the garment itself.

Techniques such as bioimpedance (BI) analysis give a better idea of total body water and fluid status. An alternating current with varying frequencies is passed between electrodes on the skin, and the resistance to this current is used to calculate water content and cell mass. In doing so, it can give a reliable and objective measure of volume status [72]. Outside of nephrology, BI has been studied in the community, where impedance across the chest has been shown to correlate with pulmonary oedema and the decompensation of heart failure, with both vests and wearable patches trialled [73,74]. Bioimpedance analysis would also incorporate an added benefit of estimating the lean body mass, which could aid in nutritional assessments. An early study using a wireless BI sensor during HD showed a greater sensitivity of BI to guide fluid removal than the traditional weight-based assessment [75]. A more recent, larger, proof-of-concept study replicated those findings, indicating that BI may be a feasible and accurate way of monitoring fluid status and guiding UF for HD patients [76]. To date, no longer-term outcomes or real-time data are available, although the ongoing BISTRO trial will hopefully aim to address some of these questions [77]. This technology appears to be on the horizon, with multiple further ongoing trials and devices in development [78], but still has limitations to overcome, such as the need to maintain a good connection between the electrodes and skin, and clearly, devices such as a vest will have compliance issues. Furthermore, BI measurements are vulnerable to extracellular fluid redistribution from postural changes, obesity and placement location of the electrodes [78]. Electrodes would also likely need to be single use or washable, which carries financial and environmental implications.

An alternative method to consider is interstitial pressure monitoring. A transducer introduced through a needle into the subcutaneous space can take pressure readings, which are shown to correlate with the total body fluid status [79]. The current technology unfortunately still involves a relatively large needle, but it is reasonable to suggest that this could become smaller and incorporated into a wearable device, such as those described for ISF electrolyte analysis. This would get around the issues with electrode connection and skin interference seen in BI but is more invasive as the trade-off, which may impact compliance. It may also have associated risks of infection, especially in the presence of oedema.

### 4.2. Laboratory Parameters

Within nephrology, POCT microsampling devices exist that can measure the serum phosphate [80], as well as serum creatinine and tacrolimus drug levels [5], with a plethora of other microfluid techniques also existing for different clinical applications [81]. However, POCT contains various disadvantages, for example, when haemolysis may erroneously elevate a potassium reading. This may trigger an unnecessary hospital attendance, whereas wearable technology may be able to circumvent this risk. A wearable device may also improve patient education and autonomy, for example, if the real-time effects on the serum potassium or phosphate levels could be seen following a diet rich in these electrolytes.

A wearable device that could provide continuous data on immunosuppression drug levels would be of great interest; compliance to drug therapy and large fluctuations in drug level are of paramount importance to not only kidney transplant patients but also in other conditions where these drugs are used, such as autoimmune diseases. This also applies to other medications—in particular, those that require dose titration according to the levels or have direct nephrotoxic effects if the therapeutic window is surpassed. Vancomycin is such an example, and drug levels have successfully been monitored via an electrochemical aptamer-based sensor that collects ISF in a paper reservoir via microneedling and correlates to serum levels during in vivo animal studies [82]. In a separate proof-of-concept animal study, machine learning was used to automatically alter the administered dose of intravenous vancomycin based on feedback from an in-dwelling intravascular sensor [83]. Although this study used invasive methods, it embodies the concept of a “clinical intelligence system” where information from non-invasive monitoring is used by artificial intelligence systems to automatically adjust the therapy [10]. Many medications already exist in a patch or pump form; therefore, a “closed-loop” system [28] that can monitor and deliver individualised, titrated therapy in a single wearable device is completely conceivable.

Whatever the wearable device that becomes available in the future, it ideally would be multifunctional. A sensor on the chest used in the LINK-HF study [79] simultaneously measured the ECG, BI, physical activity and respiratory rate, and it is likely that future devices could do this and more. A key component will be to utilise smartphone apps with wireless data collection and transfer allowing rapid communication with healthcare providers, as illustrated in Figure 4. Integrating artificial intelligence, which is able to monitor, analyse trends, predict events and automatically adjust therapy appropriately using feedback control systems, would be one step further to personalised and precision medicine and seems the inevitable next step. The use of neural networks has already been shown to outperform nephrologists when assessing HD patients’ dry weights with various clinical implications [84]. Imagine a future where a single wearable patch combined with a smartwatch could monitor electrolytes, body fluid status and biometric parameters and then deliver a titrated therapy in response, be it pharmacological or through dialysis.

Finally, the majority of studies describe the use of wearable technology in the home or an ambulatory setting. However, wearable devices outside nephrology have been shown to have a clinical utility in the inpatient setting for both diabetes control [85] and the treatment of decompensated heart failure [86], which bears consideration. Wearables may help reduce pressure on stretched resources, such as healthcare providers spending time on traditional phlebotomy, erroneous timing of samples such as tacrolimus or vancomycin levels, and reliance on traditional inpatient dialysis machines, and may help guide therapy such as more personalised diuresis. Having a less-invasive, wearable option may also be preferable for the patient, for example, those with difficult venepuncture, conservation of precious vasculature for future dialysis access or those with a needle phobia.

## 5. Challenges, Risks and Factors Associated with Successful Implementation

Wearable technologies intrigue physicians, as well as companies and their stakeholders, but it is important to consider the potential problems. One concern is whether patients, relatives and carers will perceive such devices as intrusive. Tran et al. assessed patients’ perceptions and found that few patients were ready to integrate artificial intelligence into their care, with up to one-third refusing to do so [87]. Of those that were receptive to this technology, most saw it as an adjunct rather than a replacement, with the loss of human-to-human interaction highlighted as a concern [87]. The uptake of any novel wearable will depend critically on how much the technology provides additional value to their care and how user-friendly the devices. It is also worth emphasising that perceptions around privacy vary, and a wearable device that may be welcomed by some patients could be perceived as intrusive by others. A good example may be the prospect of a wearable dialysis machine, the size and weight of devices are decreasing; however, they still remain very visible, and whilst some may find the technology convenient, others may consider it indiscrete.

Another concern is the “digital divide”, which describes the fact that parts of our patient population have access to technology and the skills to use it and others do not [88]. It is further compounded by the fact that younger and IT-savvy physicians, service developers and entrepreneurs tend to spend more time with the equally IT-literate part of their patient population than with elderly patients with limited IT skills. In some areas, the digital divide may coincide with other divides, such as those of education, wealth or ethnicity [89]. IT-literate patients are also often over-represented within patient interest groups, leading to a scenario where novel technology may only cater to a part of the population. We should make a conscious effort to also include less IT-literate patients in our considerations of wearable devices. Another important consideration is what support and patient education would be required to overcome barriers instead of just accepting them [90]. The same is true for patients with disabilities and those affected by language barriers.

Issues also exist around information governance, and uploading patient identifiable data to third-party websites is tightly regulated in most developed countries. Clinicians should aim for integration in existing electronic health record (EHR) systems [91]. These issues will become more relevant when clinicians use multiple devices and platforms concurrently, and an industry standard for telemedicine and wearable devices would help mitigate against such an uncoordinated growth of portals. Cost implications also exist, and purpose-built interfaces to link a telemedicine application into an existing EHR typically cost four-figure to low five-figure sums [3]. Finally, the “Internet of Things” opens the doors for new vectors of attack, including malware and denial of service [92], together with unprecedented access to a patient’s lifestyle, location and habits with security implications.

Potential obstacles also exist around the interpretation of data gleaned through the use of wearables. Electrolytes are particularly pertinent, and how to use continuous values that are susceptible to fluctuations needs to be clarified. Consider the use of a wearable device for potassium monitoring: Our understanding and thresholds for intervention are based on a “snapshot” of our patients’ potassium levels, i.e., at the time of phlebotomy. With a wearable device, how much of a transient potassium deviation is acceptable and for how long? Research will be required to understand continuous readings and adjust clinical pathways. The issues of big data handling and alert fatigue should also not be underestimated, and artificial intelligence and machine learning will likely be required to cope with the quantities of information gleaned by wearables [93].

Factors for the successful implementation of wearables have been described [94]. Any new wearable should address a clearly defined clinical problem, be embedded in an established pathway of care and enhance the user experience (Table 1). Financial sustainability is another issue often ignored during the early phase of implementing new technology. In this regard, not only the cost of the kit but also maintenance and repair, the supply chain, infrastructure for delivery and interface to the EHR need to be considered and funding carefully assessed and planned. Considering these aspects early on and establishing a reimbursement model are critical for long-term success [94]. Table 2 summarises the challenges and potential pitfalls of wearables.

## 6. Conclusions

It is difficult to predict how much wearables will be used in clinical practice a decade from now; developments in this area remain as unpredictable as trends in the laboratory results of some of our patients. However, we have ourselves underestimated the uptake of technology previously, sometimes spectacularly so [2]. It is therefore possible that our current view again underestimates the future developments in this field. The perceptions of patients, their relatives and other care providers also remain difficult to predict. One key factor for successful implementation will be a clear focus on a defined clinical scenario that a wearable device will address, in conjunction with the evidence of improved outcomes and/or patient experience. Integrating artificial intelligence and machine learning will likely be important further steps. Clinicians and their commercial partners should also consider very early on how their approach can be funded and how it will align with local models of care and reimbursement. With these caveats in mind, we predict that nephrology audiences will see some well-designed wearables, for example, in the management of the fluid status and electrolytes, moving from the designer stage to prêt-à-porter within the next decade.

## Figures and Tables

**Figure 1 sensors-23-01361-f001:**
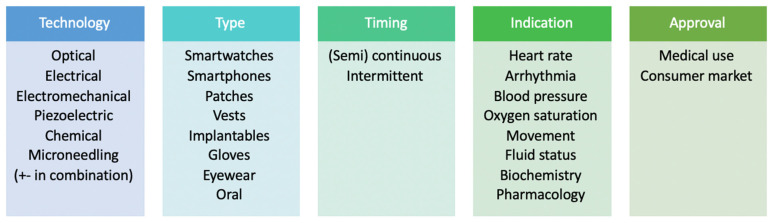
Types and categories of wearable devices used for diagnosis and monitoring in chronic kidney disease.

**Figure 2 sensors-23-01361-f002:**
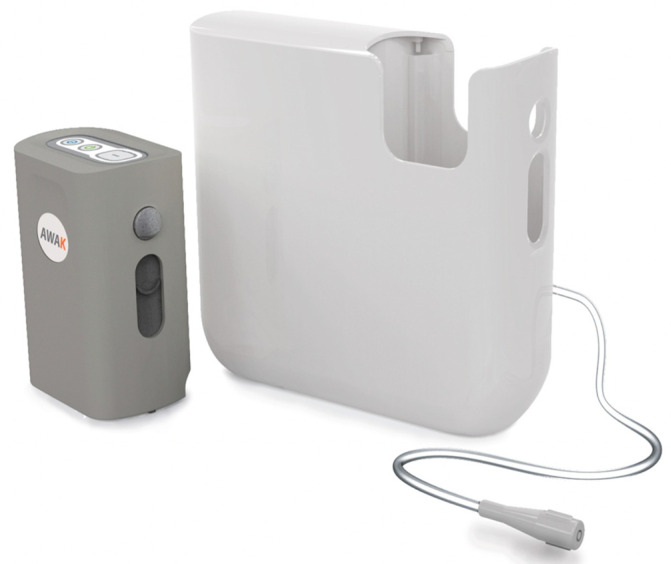
The Automated Wearable Artificial Kidney (AWAK) peritoneal dialysis device.

**Figure 3 sensors-23-01361-f003:**
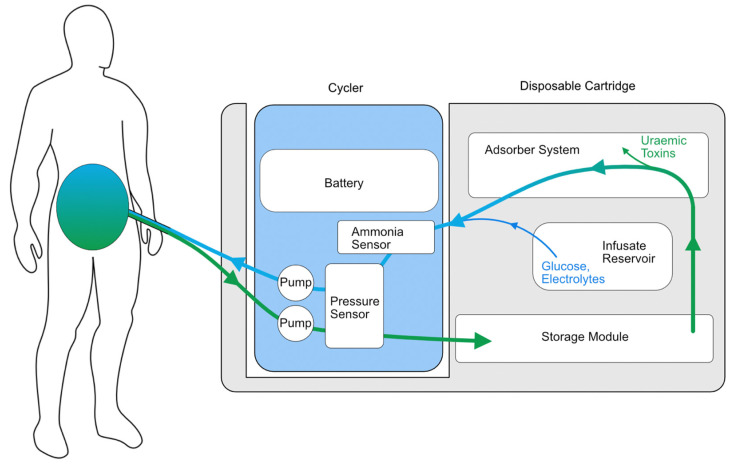
Illustrative schematic of the AWAK wearable PD device.

**Figure 4 sensors-23-01361-f004:**
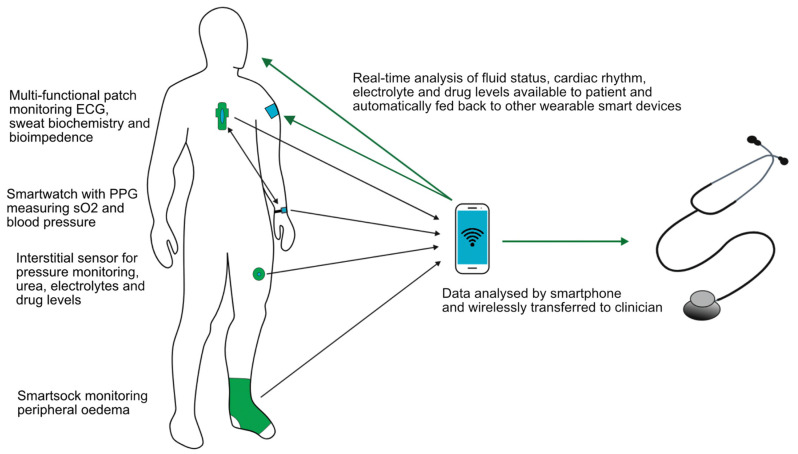
Schematic of ideal components of a multifunctional wearable device.

**Table 1 sensors-23-01361-t001:** Factors for the successful implementation of wearables in medicine (modified from [94]).

Clearly defined problem
Integrated system of healthcare delivery
Technology support
Personalized experience
Enhanced end user experience
Aligned payment and reimbursement models
Clinician champions

**Table 2 sensors-23-01361-t002:** Challenges and potential pitfalls of wearables.

Concerns around patient privacy
Digital divide
Access for patients with special needs, disabilities or where a language barrier exists
Information governance
Integration into existing electronic health records
Use of patient identifiable information on external i.e., company websites
Vulnerability to cyber attack
Large data volumes
Patient and clinician buy in, technology fatigue
Financial sustainability

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
