# Peer review of "Wearables in Nephrology: Fanciful Gadgetry or Prêt-à-Porter?"

_sensors, 2023, doi:10.3390/s23031361_

Round 1

Reviewer 1 Report

The manuscript by M. Stauss, et al. summarizes the prospects of wearable devise towards nephrological applications.  I think it clearly and concisely categorizes the physical types of wearables, their mode of operation, use-case, and target demographic.  The review initially focuses on technologies with a high readiness and then naturally transitions to concepts further from realization, but which still hold promise. Lastly, the review discusses issues and pitfall of nephrological wearables including the digital divide, information governance, and user adherence – issues which all wearables face.  Overall, I recommend publication after minor revision.

Additional Comments:

1.     Table 1 and 2 require improved formatting.

2.     As mentioned, the issues discussion largely focuses on problems faced by most wearables.  Do the authors think there are problems faced uniquely by nephrological applications? If so, it could be interesting to include this distinction.  

Author Response

Many thanks for your comments. In response to your review comments:

Point 1. Table 1 and 2 require improved formatting.

Thank you for drawing this to our attention, the tables have been adjusted and re-formatted. We have also underlined the paragraph subheadings as the original word document formatting was lost during file conversion for better ease of reading. 

Point 2. As mentioned, the issues discussion largely focuses on problems faced by most wearables. Do the authors think there are problems faced uniquely by nephrological applications? If so, it could be interesting to include this distinction.

This is a very interesting point, many thanks. The issue of how to use and interpret continuous data on electrolytes will perhaps be one of the bigger challenges for clinicians. Similarly wearable dialysis machines may be seen as intrusive or indiscrete by some. We have added these points to the discussion. Elsewhere in the manuscript we have attempted to identify where there may be specific issues within nephrology, such as using PPG in patients with multiple CKD-specific vascular abnormalities (page 3). 

Reviewer 2 Report

1.            The paper is long. You need to summarize some Sections and shorten the paper accordingly.

2.            The paper presentation is not clear. What is the objective of the conducted research? What is the novelty related scientific literature ?

3.            Several strong modifications are needed to produce an acceptable updated version.

4.            Please highlight the significance of your findings

5.            The contribution and novelty are not clear at all, the author should rewrite some section highlighting the originality and novelty proposed

6.            A final proof-reading is highly suggested, in order to correct some typos and some formatting issues.

7.            Some paragraphs are too long. Please divide them into several short paragraphs to improve the readability.

8.            Enhance the readability of the paper, in particular, transitions from section to section should be smoother.

9.            If possible, organize a summary of the Related Work section in a table to improve the quality of your research work.

10.          In the related work section, the authors cited only the previous works in the literature. The authors should also give a short comparative study between the mentioned methods.

11.          In the light of many similar works, I would also ask the authors to be more specific about how this work advances the field and why it is relevant for the scope of this journal.

12.          The author should add a related works section to introduce some recent related works

13.          More experiments, especially comparative experiments, should be involved

Author Response

Many thanks for your comments. Myself and the other authors have read them, however are unsure if some are applicable to our manuscript? For example:

Point 1 (The paper is long. You need to summarize some Sections and shorten the paper accordingly).

The suggested length by the Journal was 5500 words and at submission our manuscript was 5000, it is now slightly over this with the additions made during review. 

Point 2 (The paper presentation is not clear. What is the objective of the conducted research? What is the novelty related scientific literature ?)

Our paper was not conducted research but rather a review, therefore there are not the standard aims and objectives as there would be for an original article reporting a piece of research. We have included the contents of the review in both the abstract and introduction.

Point 4 (Please highlight the significance of your findings) and Point 5 (The contribution and novelty are not clear at all, the author should rewrite some section highlighting the originality and novelty proposed)

As mentioned above, there are no specific findings as it is not a piece of research, but rather a review giving an overview of the topic. Therefore their significance cannot be highlighted, however in our introduction we highlight why this is an important topic to be reviewed. 

Point 9 (If possible, organize a summary of the Related Work section in a table to improve the quality of your research work) and Point 10 (In the related work section, the authors cited only the previous works in the literature. The authors should also give a short comparative study between the mentioned methods).

I am not sure I follow, there is no Related Work section? 

Point 11 - In the light of many similar works, I would also ask the authors to be more specific about how this work advances the field and why it is relevant for the scope of this journal.

To our knowledge there are currently no reviews which explore wearable technology for both the diagnosis and management of CKD patients combined with wearable dialysis technology. As above, it does not present findings from a piece of individual research, and the scope of the review was agreed with the editor prior to writing. 

Point 12 (The author should add a related works section to introduce some recent related works) and Point 13 (More experiments, especially comparative experiments, should be involved)

As it is a new field there are very few studies available and we have included all of the published research. There are no comparative experiments and most technologies are still in the conception or feasibility trial stage. 

Many thanks for your comments. On behalf of the authors I hope I have clarified some of the points raised.